# Interactions between Culturable Bacteria Are Predicted by Individual Species' Growth

Einat Nestor,[a] Gal Toledano,[a] Jonathan Friedman[b]

[a]The Rachel and Selim Benin School of Computer Science and Engineering, Hebrew University, Jerusalem, Israel
[b]Institute of Environmental Sciences, Hebrew University, Rehovot, Israel

**ABSTRACT** Predicting interspecies interactions is a key challenge in microbial ecology given that interactions shape the composition and functioning of microbial communities. However, predicting microbial interactions is challenging because they can vary considerably depending on species' metabolic capabilities and environmental conditions. Here, we employ machine learning models to predict pairwise interactions between culturable bacteria based on their phylogeny, monoculture growth capabilities, and interactions with other species. We trained our models on one of the largest available pairwise interactions data set containing over 7,500 interactions between 20 species from two taxonomic groups that were cocultured in 40 different carbon environments. Our models accurately predicted both the sign (accuracy of 88%) and the strength of effects ($R^2$ of 0.87) species had on each other's growth. Encouragingly, predictions with comparable accuracy could be made even when not relying on information about interactions with other species, which are often hard to measure. However, species' monoculture growth was essential to the model, as predictions based solely on species' phylogeny and inferred metabolic capabilities were significantly less accurate. These results bring us one step closer to a predictive understanding of microbial communities, which is essential for engineering beneficial microbial consortia.

**IMPORTANCE** In order to understand the function and structure of microbial communities, one must know all pairwise interactions that occur between the different species within the community, as these interactions shape the community's structure and functioning. However, measuring all pairwise interactions can be an extremely difficult task especially when dealing with big complex communities. Because of that, predicting interspecies interactions is a key challenge in microbial ecology. Here, we use machine learning models in order to accurately predict the type and strength of interactions. We trained our models on one of the largest available pairwise interactions data set, containing over 7,500 interactions between 20 different species that were cocultured in 40 different environments. Our results show that, in general, accurate predictions can be made, and that the ability of each species to grow on its own in the given environment contributes the most to predictions. Being able to predict microbial interactions would put us one step closer to predicting the functionality of microbial communities and to rationally microbiome engineering.

**KEYWORDS** computational biology, machine learning, mathematical modeling, microbial ecology, microbial interactions, synthetic microbial communities

Address correspondence to Jonathan Friedman, yonatan.friedman@mail.huji.ac.il.

The authors declare no conflict of interest.

Microbes are key participants in various processes, ranging from the health of humans (1), animals, and plants (2) to global biogeochemicals cycles (3). The impact of microbes, however, is usually not due to a single species but rather caused by diverse communities of interacting species (4). Therefore, the mechanisms by which microbial species promote or hinder each other's growth has been studied extensively (5). For example, negative effects can occur due to resource competition or secretion of antimicrobials (6, 7), whereas positive ones may occur due to crossfeeding (8) of metabolites, such as amino acids (9).

10.1128/msystems.00836-22 **1**

Predicting interspecific interactions is necessary to understand a community's properties, as they are expected to be shaped by interactions within the community (10–13). Indeed, pairwise interactions have been shown to be predictive of the structure and function of various simplified microbial communities (14–18). However, it can be extremely challenging to directly measure all pairwise interactions in a community or to infer them from sequencing data (19–21). An alternative approach, which is likely essential for species-rich communities or ones comprised of fastidious species, is developing methodologies for predicting how microbes affect each other's growth.

Metabolic modeling and genome-based models have been commonly used to predict microbial interactions (22–24). These approaches predict interactions by considering the overlap and complementarity between species' metabolic capabilities and/or their resource consumption and secretion (25, 26). These approaches are appealing because they rely primarily on genomic information. However, their performance depends on the availability of well-annotated genomes, and they typically do not account for nonmetabolic interaction modalities, such as the secretion of antibiotics or pH modifications (27).

Another promising approach to interaction prediction is the use of machine learning models (28). The use of supervised and unsupervised machine learning algorithms has increased in the past few years in many biological fields, including microbiology (29, 30). Previous works have managed to show that microbial community composition can be predicted using deep learning (31, 32). In addition, the use of supervised machine learning tools to accurately predict the sign of microbial interactions (positive or negative) based on genomic data and inferred metabolic pathways has recently been demonstrated (28). While the latter results for bacterial interaction prediction are promising, they are restricted to engineered auxotrophic species, *in silico* simulations, and a handful of soil species in a single environment. Given that interactions vary significantly between species and can drastically change across environments, even between the same species, it is still not clear to what extent machine learning tools can predict interactions between nonengineered species across a range of nutrient conditions.

Here, we assess the ability of machine learning tools to predict microbial interactions using one of the largest data sets of experimentally validated microbial interactions. This data set contains all pairwise interactions among 20 different soil bacteria from two taxonomic groups that were cultured in 40 different media, each containing a single carbon source or a mixture of all carbon sources. Combined with phylogenetic information and phenotypic features, which were created from the data set, machine learning models were able to accurately predict both the sign and the strength of pairwise interspecific interactions in this data set.

## RESULTS

In order to predict how species affect each other's growth, we have used additional information, beyond the interspecific interactions, regarding the species' phylogeny and their monoculture yield in each of the 40 carbon environments. The growth yield of all species in monoculture and in coculture with each other species in each carbon environment was measured using the kChip combinatorial screening platform (33). The one-way effect of one species on another in a given environment was quantified as the log ratio of the affected species' growth yield in coculture and in monoculture in that environment (see Materials and Methods). Information regarding each species' phylogeny and metabolic capabilities were included as features based on the species phylogenetic or monoculture growth profile similarity (represented as the first two principal components, abbreviated as PCs, of the phylogenetic distance matrix for each species or as the first four PCs of the monoculture growth distance matrix; see Materials and Methods and Fig. S1). We have first used this large data set to train machine learning models to predict either the sign (positive/negative) or strength of one-way effects of one species on another's growth yield (see Fig. 1; Tables S1, S2 at 10.6084/m9 .figshare.21856512, 10.6084/m9.figshare.21856557, respectively).

**Machine learning algorithms predicted both the sign and the strength of one-way growth yield effects well.** We evaluated the predictive ability of several machine learning algorithms and found that tree-based models performed best for predicting both effect sign and strength (in particular, XGBoost performed best for both sign and strength, Fig. S2).

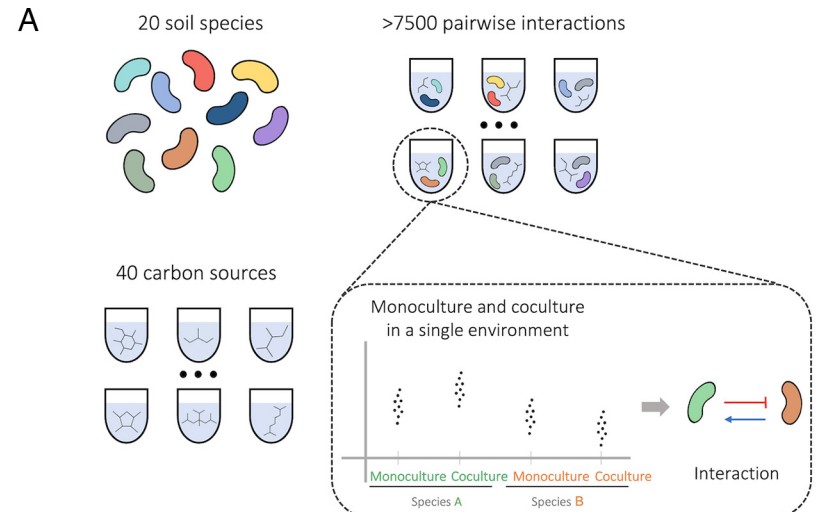

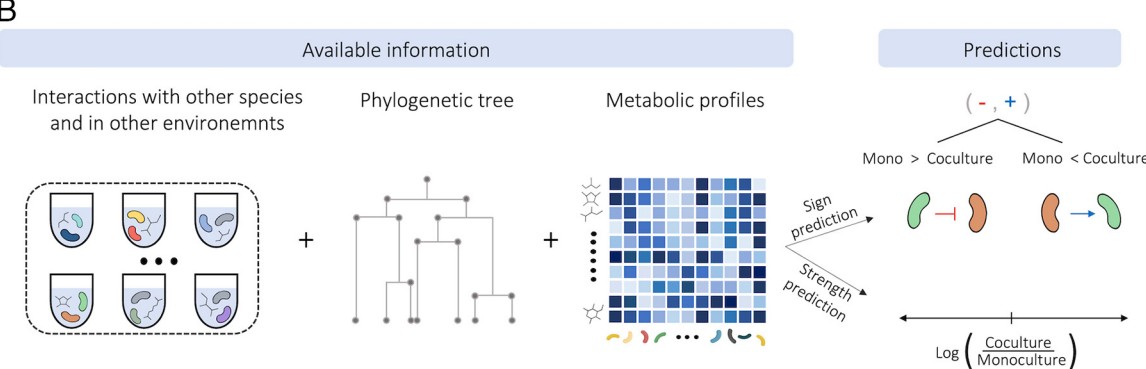

**FIG 1** An illustration of the data used to quantify and predict bacterial interactions. (A) All pairwise interactions among 20 different soil bacteria from two taxonomic groups, cocultured in 40 different carbon sources were measured. (B) Information regarding species' phylogeny and metabolic capabilities were created from the data set. Machine learning models were trained on the data, in order to predict the sign and strength of effects of one species on the growth of another.

Briefly, XGboost uses an ensemble of decision trees, where each tree makes predictions by iteratively splitting the data based on individual features. XGboost implements a form of gradient boosting such that each new tree focuses on the samples where the previous trees had the highest error rates. XGboost is widely used because it provides accuracy and efficiency even on large data sets containing many features (34). The performance of these models was also superior to that of null models that always predict the most frequent sign/ average effect strength of train set, and threshold models that use a predefined threshold of a single feature (e.g., predict a negative effect if the metabolic distance between the interacting species is above a threshold, or the monoculture growth of the affected strain is above a threshold). These results confirm that machine learning models can predict both effect sign and strength better than models with a simple decision role.

Tree-based models accurately predicted both the sign and strength of one-way effects. Quantitative predictions of effect strength achieved a normalized root-mean-square error (NRMSE) of 0.36 and $R^2$ of 0.87 on the validation set (Fig. 2A). Qualitative predictions of effect sign had an out-of-sample accuracy of 0.88 as well as high precision (0.67), recall (0.8), and Mathews correlation coefficient (0.66), which accounts for the fact that our data are imbalanced with 76% negative effects (Fig. 2B; Fig. S3A). Most errors in effect sign prediction (3.8% false positives, 7.8% false negative) occurred for effects whose strength was close to 0 (Fig. S3B), indicating that the model was able to distinguish well between effects that were strongly negative or positive, but had difficulties in classifying weaker ones. Moreover, it appears that predictions involving positive effects were less accurate: true negative effects

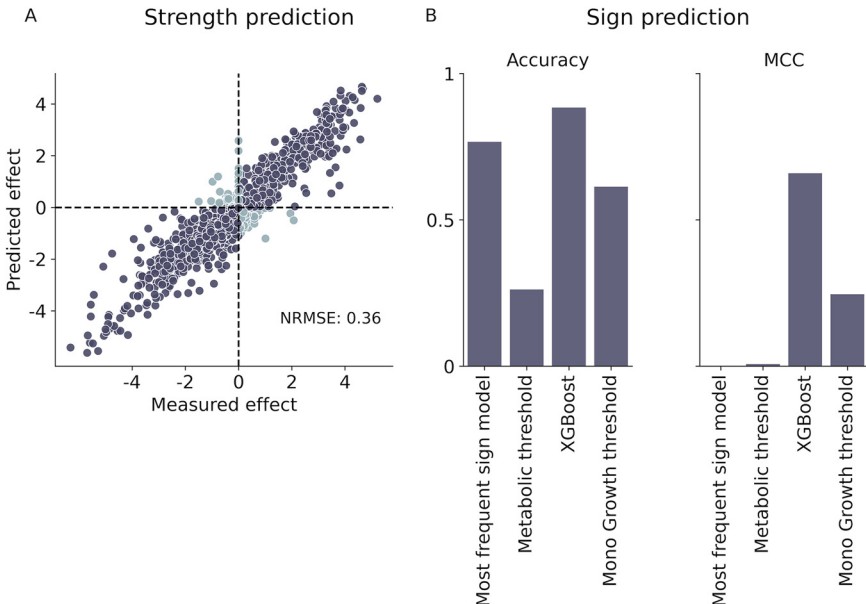

**FIG 2** Tree-based machine learning models accurately predicted the sign and strength of one-way growth yield effects. (A) Measured effect versus predicted effect for each species pair in the validation set for the best quantitative model, XGboost (Materials and Methods). Dark gray indicates samples with the same sign in actual effect and predicted effect while light blue indicates samples with different signs. (B) Comparison of Accuracy and Matthews correlation coefficient for the best qualitative model (XGBoost classifier), null model (assign common effect sign in train set) and two simple one feature decision role models: metabolic distance threshold model and monoculture growth of affected strain threshold model (Materials and Methods). Other metrics (recall and precision) are in Fig. S3.

were correctly classified more frequently than positive ones (95% true negative rate versus 66% true positive rate; Fig. S4A) and effects classified as negative were more likely to be true than effects classified as positive (90% negative predictive value versus 80% positive predictive value; Fig. S4A). In addition, while naive models achieved similar accuracy as the XGBoost, they performed very poorly in all other matrices (Fig. 2B; Fig. S3A), strengthening the conclusion that simple decision roles offered little predictive power.

**Monoculture growth yield is the most predictive feature.** We further analyzed the contribution of each feature to the performance of the models using SHapley Additive exPlantations (SHAP), a game theory approach that measures the contribution of each feature to the total prediction of the model (35, 36). How well both species can grow in monoculture in the carbon environment in which they were interacting had the strongest influence on the prediction of both effect sign and strength (Fig. 3; Fig. S5). SHAP analysis indicates that higher monoculture growth yield of the affected species leads to a stronger negative contribution to the model's output. In other words, species that grow better in monoculture tend to be more negatively affected by the presence of additional species. This is consistent with previous findings that monoculture yields shape pairwise interactions (33). Surprisingly, using information regarding species' predicted metabolic pathways, which were previously shown to be predictive of interactions (28), instead of information regarding monoculture growth did not improve the predictive ability over a model that only used the species' phylogeny (Fig. S6).

While tree-based models offered improved predictive power compared to simpler models, they relied on having information regarding each species' monoculture growth and interactions with many other species in each carbon environment. Obtaining such information can be a laborious and challenging task, especially for species that are hard to culture under laboratory conditions. Therefore, we next studied how accurately we can predict the sign and strength of one-way effects when only partial information is available. To do so, we trained new models with only partial information regarding one of the species and compared the accuracy of prediction to those of the models trained using all the data.

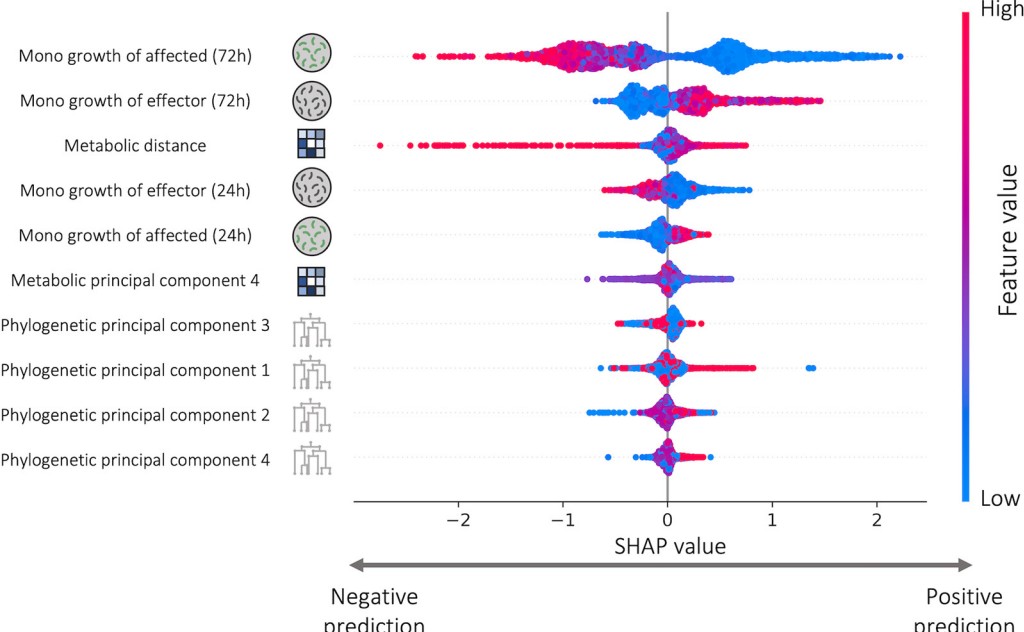

**FIG 3** Monoculture growth is the most important feature for predicting the strength of effect one species has on another's growth. Top 10 most important features in the best performing strength prediction model (XGboost). High SHAP values indicate positive influence on the predicted effect, and low values indicate negative influence on the predicted effect. *y* axis represents the different features, sorted in descending order according to their contribution to the model which is defined by the mean absolute SHAP value of all simulations for that feature. Each dot represents a simulation of the model with a single change at a single feature value. Note that the different colors do not represent the SHAP value, but the value of the feature. Similar contribution of monoculture growth features appeared also in sign predictions (Fig. S5A).

First, we evaluated our ability to predict interactions involving species for which we have monoculture growth data but no coculture data by removing a species from the training set. Next, we evaluated the accuracy of prediction when neither monoculture nor coculture data are available by removing a species from the training set and removing features related to monoculture growth from both the training and testing sets. In the latter case, predictions are based only on phylogenetic information. Lastly, we included a naive "phylogenetic copy" model where the sign or strength of the effect is assigned to be identical to those that involve the phylogenetically closest species in the same carbon environment (Fig. 4A; Materials and Methods).

The accuracy of predicting the sign and strength of one-way effects depended strongly on the availability of monoculture growth data, but not on coculture data (Fig. 4A and C). The lack of coculture data involving a given species increased the median prediction error (quantified using the NRMSE) by 0.15 (from 0.42 to 0.57), whereas removing monoculture data increased the median error by 0.25 (from 0.57 to 0.82). Moreover, when monoculture growth data are not available, prediction quality was similar to that of the simple "phylogenetic copy" model, which only requires the phylogenetic distance matrix (median RMSE values of 0.82 and 0.77).

In a similar way, we evaluate our ability to predict interactions that occur in a carbon environment for which we have only partial information. First, we removed a carbon environment from the training set. Next, we also removed features related to monoculture growth from both the training and testing sets. Lastly, we included a naive "metabolic copy" model where the sign or strength of effect is assigned to be identical to that of the same species in the metabolically closest carbon environment (Fig. 4B; Materials and Methods).

We again found that prediction accuracy depended more strongly on the availability of monoculture growth data than on coculture data (Fig. 4D), although overall predictions were less accurate. The lack of coculture data involving a given environment increased the median prediction error by 0.21 (from 0.43 to 0.64), whereas removing monoculture data

**A** — With partial information about species

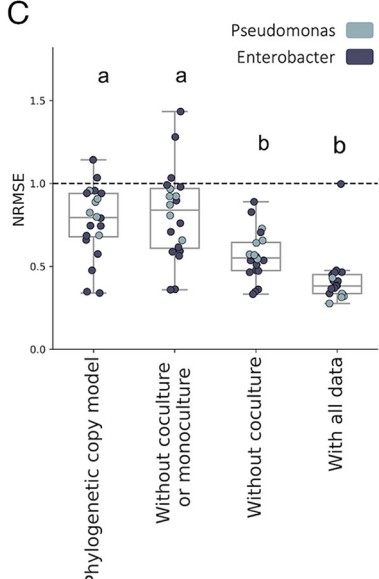

| | Focal species in training set | Monoculture growth | Carbon PCs | Metabolic distance | Phylogenetic PCs |
|---|---|---|---|---|---|
| With all data | + | + | + | + | + |
| Without Coculture | - | + | + | + | + |
| Without Coculture or Monoculture | - | - | + | - | + |
| Phylogenetic copy model | - | - | - | - | + |
| Average strength | - | - | - | - | - |

**B** — With partial information about Environments

| | Focal environment in training set | Monoculture growth | Carbon PCs | Metabolic distance | Phylogenetic PCs |
|---|---|---|---|---|---|
| With all data | + | + | + | + | + |
| Without Coculture | - | + | + | + | + |
| Without Coculture or Monoculture | - | - | + | - | + |
| Metabolic copy model | - | - | + | - | - |
| Average strength | - | - | - | - | - |

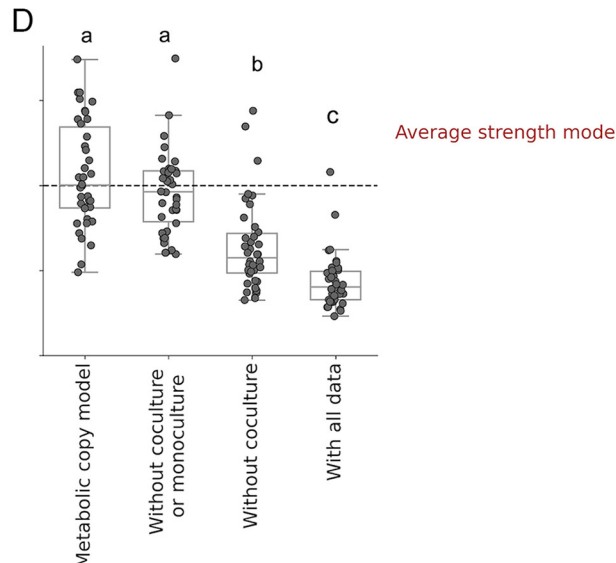

**FIG 4** The accuracy of predicting one-way effect depends strongly on the availability of monoculture growth data, but not on coculture data. A. A summary of the 4 compared models and the information used by each model (full or partial species' information). Partial information (left to right): Coculture, Monoculture (of both species), carbon PCs, metabolic distance and phylogenetic PCs. The models differ by the features they are trained on and their test set (see Methods). B. A summary of the 4 compared models and the information used by each model (full or partial environments' information). Partial information (left to right): Coculture, Monoculture (of both species), carbon PCs, metabolic distance and phylogenetic PCs. The models differ by the features they are trained on and their test set (see Methods). C. Comparison of NRMSE (normalized root mean square error) of several partial quantitative models, on "uncultured" strain. Each dot represents the NRMSE of a different strain that was excluded from the train set and was included in the test set. The models are the same as in A. Dashed line represents the performance of a null model (average effect in the train set). D. Comparison of NRMSE (normalized root mean square error) of several partial quantitative models, on "uncultured" carbon environment. Each dot represents the NRMSE of a different carbon source that was excluded from the train set and was included in the test set. The models are the same as in B. Dashed line represents the performance of a null model (average effect in the train set). Qualitatively similar results were found also for predictions of effect sign (Fig. S7). P-values for subplots C and D were calculated using Tukey-HSD test (see Table S3 at 10.6084/m9.figshare.21856578 for additional information regarding the P-values).

increased the median error by 0.32 (from 0.64 to 0.96). The same pattern of improvement for sign predictions occurs when only partial information is available regarding the interacting species or the environment (Fig. S7). These results indicate that if a species' monoculture growth in a given carbon environment is known, growth effects involving that species can be well predicted given other species interactions in the same environment, or the same species' interactions in other environments.

**Accuracy of "phylogenetic copy" model was higher for closely related species.** Given that our best option for predicting interactions involving "uncultured" species (ones for which we have no monoculture or coculture data) was the simple "phylogenetic copy" model, we next examined how the phylogenetic distance from the "copied" species (for which interaction information is available) affects the prediction quality. As expected, the prediction accuracy and distance from "copied" species were significantly positively correlated (Fig. 5; Pearson

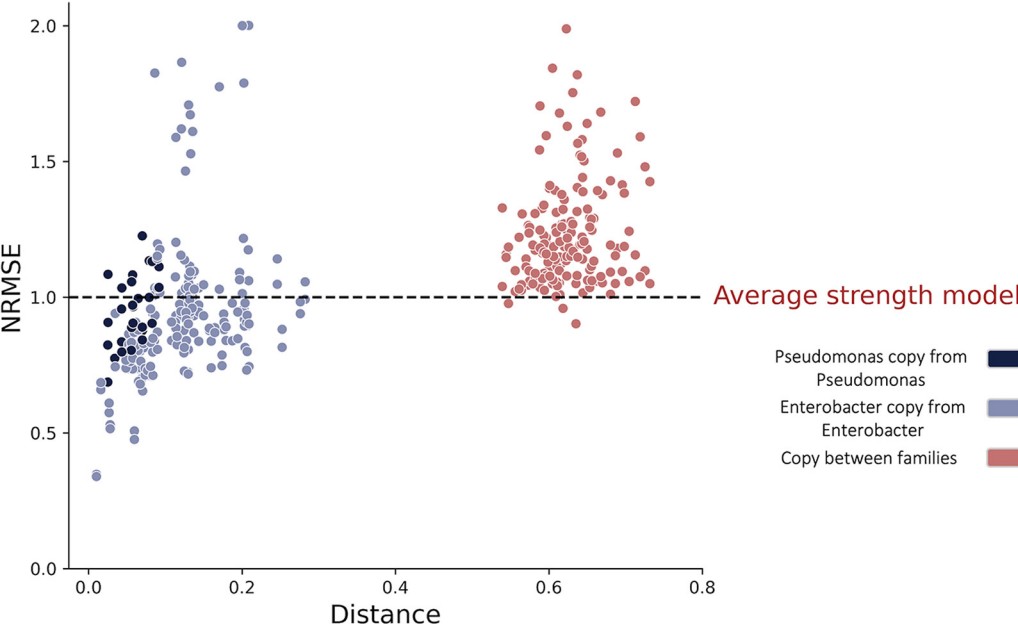

**FIG 5** A simple phylogenetic copy model can provide predictive power when based on closely related species. Each dot represents a prediction for a single species, using one of the other 19 species as the copied species (interaction of the copied species in the same carbon source). Different colors represent the distance category: the copied strain is from the same taxonomic family (dark blue, light blue) or form a different taxonomic family (red). Other metrics for sign predictions (Matthews correlation coefficient, accuracy, recall, and precision) are found in Fig. S8.

correlation coefficient 0.56, $P$ value <0.001). In other words, strains that are phylogenetically similar to the uncultured strain will be better predictors of the uncultured strain at various carbon sources whereas the greater the distance, the worse the predictions gets. However, poor prediction accuracy, lower than that achieved using the average effect strength, sometimes occurs even when copying interactions from species within the same family, and prediction accuracy varied between families. These results indicate that interactions tend to be conserved between closely related species, but the extent of conservation may vary between taxonomic groups.

**Combining one-way effect predictions is as accurate as jointly predicting two-way interactions.** Lastly, we studied how well we can predict two-way interactions that comprise both one-way effects of the interacting species on one another. We predicted two-way interactions using the same best-performing tree-based models that were used for predicting one-way effects (retrained for multilabel output; see Materials and Methods). Similar to one-way effects, we quantified the accuracy of qualitative predictions of interaction type: competition (−, −), mutualism (+, +) and parasitism (−, +) and of quantitative predictions of interaction strength (Materials and Methods).

Surprisingly, jointly predicting two-way interactions was not more accurate than combining the independent predictions of two one-way effects (Fig. 6A and B; Fig. S9). As positive one-way effects are harder to predict, mutualisms (+/+) are particularly challenging and are more often classified as parasitisms (+/−) than as mutualisms (48% versus 33%; Fig. S4B). To better understand this finding, we quantified the dependence between reciprocal effects between a pair of species using maximal information coefficient (37) (MIC), a metric for capturing general dependencies between variables that ranges from 0 (independent) to 1 (fully dependent). Reciprocal effects between a pair of species were only weakly dependent on one another (MIC = 0.16), indicating that knowing how one species affects another is not very predictive of the reciprocal effect.

In addition, we trained the same one-way strength model used to predict one-way effect, but with the reciprocal effect as an additional feature. Adding the reciprocal effect had little effect on prediction accuracy (NRMSE decrease of 0.01; Fig. 6C) and the reciprocal effect contributed little to predictions (Fig. S10). In other words, knowing the other species' effect does

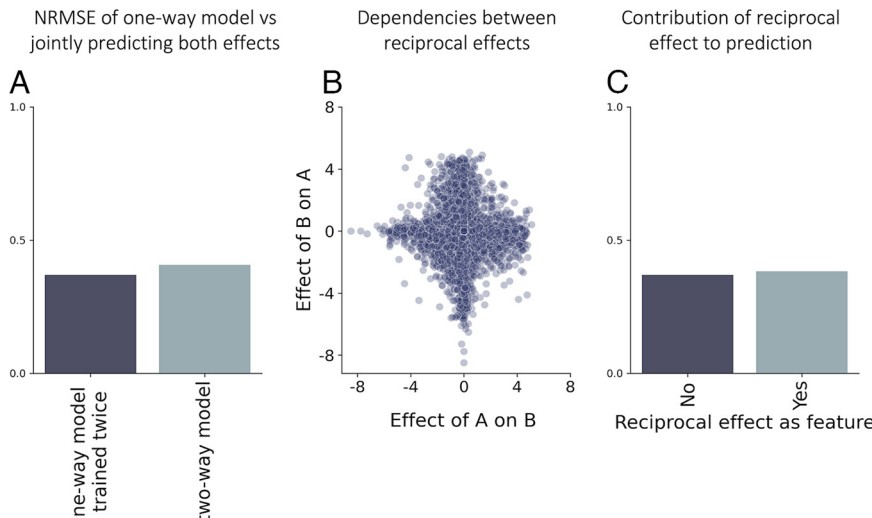

FIG 6 Reciprocal effects are weakly dependent on one another and therefore do not improve the accuracy of interaction predictions. (A) NRMSE for using one-way strength model to predict two-way effects, and with multilabel strength model. (B) The relationship between the one-interaction effects. Each dot is a single interacting species, where the $x$ axis and the $y$ axis are the effects of one species on the other. (C) NRMSE for one-way strength model without the reciprocal effect as a feature, and with the reciprocal effect as a feature.

not add any helpful information, as this information is redundant when other features (monoculture growth yields, metabolic PC, and phylogenetic PC) are available.

## DISCUSSION

Microbial interactions can help predict the properties of microbial communities but are challenging to measure (19–21). Here, we demonstrate that tree-based machine learning models can accurately predict the sign and, more importantly, the strength of bacterial interactions. These predictions were based on the species' phylogeny as well as on phenotypic features which are extracted from the monoculture growth yields of the species in various carbon courses.

The ability of the affected species to grow in monoculture in a given carbon environment was the feature that contributes the most to prediction. This is consistent with the fact that methods such as metabolic modeling, which are primarily based on genomic information, require refinement using experimental measurements such as monoculture growth (38–42). Consistent with previous findings, species that grow well in monoculture are predicted to be more negatively affected by coculturing with other species, and to affect other species more positively. While prediction accuracy depended strongly on the availability of a species' monoculture growth data, it was less sensitive to the removal for coculture data. This is encouraging, because it indicates that the number of measurements allowing accurate interaction prediction scales linearly, rather than quadratically with the number of species.

In the absence of monoculture growth data, a simple phylogenetic copy model, which is intuitive and easy to create, offered some predictive power. In this model, interactions between a pair of species were predicted to be identical to those that occur between closely related species in the same environment. This indicates that bacterial interactions are to some extent phylogenetically conserved, at least within the two families analyzed here, and that known interactions may be informative regarding the interactions between other, closely related, species for which no growth data are available.

In contrast, predicting interactions in new carbon environments was significantly less accurate. Predicting that the interaction between a pair of species was identical to the interaction of the same pair in the most "similar" carbon environment was not accurate. This poor accuracy may be due to the fact that carbon sources are not clustered into distinct groups based on the species' growth abilities (like species are clustered according to the phylogenetic tree; Fig. S1). More in-depth research is needed in order to best use

information from one environment to make predictions regarding another environment, which may improve interaction predictions, especially in "new" carbon environments (43).

Surprisingly, models that were trained using information regarding each species' inferred metabolic pathways did not achieve higher prediction accuracy than models that used only phylogenetic information (Fig. S6). However, the metabolic pathways were inferred from the 16S sequences using picrust2 (44), rather than from well-annotated whole-genome sequences. Therefore, it is possible that the addition of metabolic pathways that are constructed from whole-genome sequences will improve the performance of the models, improve prediction accuracy for uncultured species, and offer insights regarding the mechanistic basis underlying bacterial interactions.

While our results demonstrate that bacterial interactions are predictable under simple laboratory conditions, it is still not clear to what extent this predictability extends to nonlaboratory settings and to natural communities. First, many microbes are hard to isolate and culture in the lab (45, 46), and therefore monoculture growth yield will typically not be available for many environmental microbes. Because monoculture growth yield was the most informative feature for interaction predictions, not having this information would likely significantly reduce predictability. Moreover, the number of available resources natural communities are exposed to is larger than what our models were trained on, which were simple environments containing a single carbon source. This may make predicting interactions more challenging as species may occupy different niches and both grow well in monoculture without negatively influencing each other's growth. Lastly, our model predicts pairwise interactions, and does not account for the presence of "higher order" interactions (47). Therefore, even if our model accurately predicts the interactions between all species pairs of a natural community, the presence of additional species in the environment may modify these pairwise interactions.

Predicting pairwise interspecific interactions is crucial for understanding the structure, stability, and function of microbial communities. Here, we demonstrate that tree-based machine learning models can be used for accurately predicting interactions of different species within the same taxonomic group or between different taxonomic groups, in a relatively large set of conditions (40 different carbon environments). Further work is needed in order to test the ability of this approach to predict interactions between more diverse taxonomic groups, and in more complex situations involving multiple species and nutrients. Being able to predict microbial interactions would put us one step closer to predicting the functionality of a microbial communities and to rationally microbiome engineering.

## MATERIALS AND METHODS

**Data.** The data set contains over 7,500 pairwise interactions involving 20 species from two taxonomic groups in 40 different carbon environments (see previous work [33]), as well as the monoculture growth yield of all species in all carbon environments. Here, ~7% of all possible combinations of species and carbon environments were not included in our analysis because they were represented by less than three replicates in the original experimental data set (the specific missing combinations are listed in Table S4 at 10.6084/m9.figshare.21856581). Briefly, species' growth in mono- and coculture was assayed using the kChip platform, a high-throughput nanodroplet-based platform for combinatorial screening (48). These data were used to calculate the growth effect of one species on another as the log ratio of the affected species growth in coculture and in monoculture. Lastly, pairwise interactions are given by both the effect of species A on B and the effect of species B on A (33).

**Features creation.** We created features representing species' phylogeny based on a previously published phylogenetic distance matrix of the 20 species (33). We performed principal-component analysis (PCA) on this matrix and used the first two principal components, which capture >95% of the variance, as features. Features that represent the carbon environments were based on the species' metabolic profiles, where the metabolic profile of each carbon environment is the monoculture growth yields of the 20 species. We performed PCA on the metabolic profile matrix and used the first four principal components, which capture >90% of the variance, as features (presented in Fig. S1). These features represent each carbon environment according to similarities in monoculture growth yields of the different species. In addition, we included a metabolic distance feature, which we calculated as the Euclidean distance between the monoculture-growth yields profiles of each pair of interacting species.

**Model training.** First, the data were split into two groups: train and test set (80% and 20%, respectively). Then, the hyperparameters of each model were tuned by performing 5-fold cross-validation on the train set and choosing the parameter values that resulted in the best performance (highest MCC for qualitative predictions, lowest RMSE for quantitative predictions). For each hyperparameter, 2,500 values were sampled uniformly from a given range, presented in Table S5 (at 10.6084/m9.figshare.21856575). The models which were

used for qualitative predictions are random forest classifier, logistic regression, K nearest neighbors classifier, and XGBoost classifier. The models which were used for quantitative predictions are: Random forest regressor, XGboost, linear regression and K nearest neighbors regressor. All models were used from scikit-learn open-source package (python). Hyperparameter tuning was made using RandomGridSearch (scikit-learn 1.0.1).

**Naive models.** In addition to machine learning models, we evaluated the performance of several simple prediction role models:

1. Null models: predict the effect sign to be the most frequent sign in the training set and the effect strength to be the average interaction strength in the training set.
2. Threshold models (for effect sign only): predict the effect sign based on whether the value of a single feature exceeds a threshold value. The threshold value was set to be the one that maximized accuracy in the training set. Two threshold models were created, one based on the metabolic distance between a pair of species and a second model based on the monoculture growth yield of the affected species.

**Models trained using partial information.** In order to evaluate our ability to predict interactions involving species or carbon environments for which only partial information is available, we created the following:

**Partial information regarding a species.** For each of the species, a different test set was created containing only the species interactions, excluding all the interactions involving the species from the train set. For each species excluded from the training set, three machine learning models were trained using different sets of features:

1. Without coculture, but with monoculture growth measurements and phylogenetic features.
2. Without coculture or monoculture growth measurements, but with phylogenetic features.
3. Only with phylogenetic features.

Additionally, a simple decision role model was evaluated:

4. Phylogenetic copy model: copies the interaction (sign or strength) of the phylogenetically closest (according to the phylogenetic distance) strain in the same carbon environment, when interacting with the same partner.

Overall, $4 \times 20 \times 2$ (four types of models, 20 species, and two types of prediction) models were trained and compared.

**Partial information regarding a carbon environment.** For each of the carbon environments, a different test set was created containing only the interactions occurring in that environment, excluding all the interactions in that environment from the train set. For each carbon environment excluded from the training set, three machine learning models were trained using different sets of features:

1. Without coculture (in the specific environment), but with monoculture growth measurements and phylogenetic features.
2. Without coculture or monoculture growth measurements (in the specific environment), but with phylogenetic features.
3. Only with phylogenetic features.

Additionally, a simple decision role model was evaluated:

4. Metabolic distance model: copied the interaction (sign or strength) in the most similar carbon environment (according to the Euclidean distance of the environments' metabolic profiles).

As the monoculture growth yields were used for creating the metabolic representation of the different carbon environment, the metabolic representation of the carbon environment excluded from the training set was generated using the PCA of the other carbon environment. Overall, $4 \times 40 \times 2$ (four types of models, 40 carbon environments, and two types of prediction) models were trained and compared.

**Model performance evaluation.** The performance of models predicting effect sign was evaluated using Matthews correlation coefficient, which accounts for the fact that negative interactions are more frequent in our data set (73%). The performance of models predicting effect strength was evaluated using normalized RMSE (NRMSE), defined as the RMSE divided by the standard deviation of the observed effects in the test set.

**Two-way interactions prediction.** A two way interaction (between species A and B) is composed of a pair of reciprocal effects (effect of B on A, effect of A on B). There are two ways to predict two-way growth effect:

1. Train the one-way effect model and predict each of the two reciprocal effects independently.
2. Train a two-way model with multi label output (each prediction is in the form of [effect of B on A, effect of A on B]) and jointly predict the two-way interaction.

**Data availability.** Code and data are available at https://github.com/einatnestor/Microbial-interaction-prediction.

## SUPPLEMENTAL MATERIAL

Supplemental material is available online only.
**FIG S1**, TIF file, 10.7 MB.
**FIG S2**, TIF file, 13.6 MB.
**FIG S3**, TIF file, 13.8 MB.
**FIG S4**, TIF file, 13.5 MB.
**FIG S5**, TIF file, 13.6 MB.

**FIG S6**, TIF file, 12.1 MB.
**FIG S7**, TIF file, 13.1 MB.
**FIG S8**, TIF file, 13.2 MB.
**FIG S9**, TIF file, 13.3 MB.
**FIG S10**, TIF file, 12 MB.

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
