## [Reviewer comments · mSystems]

Interactions between culturable bacteria are predicted by individual species' growth

Jonathan Friedman, Einat Nestor, and Gal Toledano

Corresponding Author(s): Jonathan Friedman, Hebrew University of Jerusalem

Review Timeline:

Submission Date:	September 5, 2022
Editorial Decision:	October 31, 2022
Revision Received:	January 11, 2023
Accepted:	January 30, 2023

Editor: Kiran Patil

Reviewer(s): The reviewers have opted to remain anonymous.

Transaction Report:

DOI: <https://doi.org/10.1128/msystems.00836-22>

October 31, 2022

Dr. Yonatan Friedman
Hebrew University
Israel

Re: mSystems00836-22 (Interactions between culturable bacteria are predicted by individual species' growth)

Dear Dr. Yonatan Friedman:

Thank you for submitting your manuscript to mSystems. We have completed our review and I am pleased to inform you that, in principle, we expect to accept it for publication in mSystems. However, acceptance will not be final until you have adequately addressed the reviewer comments.

Please add a brief discussion on the limitations of the approach when concerning natural communities whereby only a limited subset of species can be cultured, and potential artefacts of laboratory cultivations.

Preparing Revision Guidelines

Sincerely,

Kiran Patil

Editor, mSystems

Journals Department
Reviewer comments:

Reviewer #1 (Comments for the Author):

The work by Nestor et al. uses a machine learning model to predict interspecies interactions from bacterial monoculture data, among other organism-specific traits. The authors leverage an extensive set of pairwise and monoculture data to characterize their model, as well as to test the contribution of different data sources on its performance. A thorough comparison with additional models is also presented. Overall, this is a clearly written and timely article that adds to the growing body of literature on machine learning methods for predicting microbiome behavior. Importantly, it also underscores how monoculture growth data is essential for correct predictions of interactions. There are a few points that in my view require further clarification, which are outlined below along with some suggestions on how the manuscript may be improved.

1. The authors use outputs from principal component analyses as features in their model (introduced in lines 77 and 349). I believe the authors are referring to Figure S9, though this figure is not referenced anywhere in the manuscript.
2. The authors contextualize their work by discussing the use of metabolic models (among other methods) and that these methods rely solely on genomic information (beginning at Line 41). While many studies indeed use models generated automatically from genome annotations, it is known that experimental curation is an important step in generating reliable quantitative predictions (PMID: 20057383), and has been applied extensively in model generation (PMID: 31391098, 28266498, 29692801, 21483480). As this is in line with the authors' observations on the essentiality of species monoculture growth, I would recommend rephrasing this statement.
3. Line 71: "in order to predict how species affect each other:" please specify what is being affected.
4. Line 73: Please change "growth" to "growth yield."
5. Line 79: Please change to "principal components."
6. Line 96: For a general audience, a description of tree-based models (and specifically of ensemble models like XGBoost) would be very useful.
7. Line 110: "most errors in sign prediction occurred for effects whose strength was close to 0." This is very interesting - were there inconsistencies in interaction sign for these examples in the original dataset? Are there other factors generally associated with these weaker interactions?
8. Line 139: Please provide citations for "which were previously shown to be predictive of interactions."
9. Figure 4: A calculation of statistical significance between the NMRSE distributions would be useful in more fully comparing each model's performance.
10. The model's predictions are based on species' growth yields in mono- and co-culture. While this is appropriate for this experimental system that contained limiting amounts of resources, natural microbiomes are open systems subject to nutrient cycling and periodic resource replenishment. I understand this study is framed as a proof of concept, but it would be valuable for the authors to comment on whether they might expect other ecologically relevant quantities (e.g., growth rate) to have similar degrees of predictive power in a non-laboratory setting. This could help guide the selection of features for future models intended for more complex microbiomes.
11. Line 335: For those not familiar with the original dataset, please clarify the degree to which each species was represented in the mono- and co-culture datasets.
12. Line 361: "highest accuracy for qualitative predictions" - is this appropriate given the imbalanced nature of the dataset (i.e. more negative interactions)?
13. Figure S2B: The total N should be specified or the figure should be normalized to the total number of effects.

Reviewer #2 (Comments for the Author):

The manuscript "Interactions between culturable bacteria are predicted by individual species growth" by Nestor et al aims at building a predictive model for pairwise interactions between bacteria. The predictive model is based on a rather unique

collection of 7500 experimental measurements of pairwise growth of 20 species grown on various carbon sources that was built as part of an earlier study (Kehe et al 2021). The current study is a natural continuation, using this data as a training set for a predictive model. The authors first demonstrate the performances of their model and then analyze the contribution of each of the different features on the performance. The importance of this work is in providing a guideline for the design of strategies for the culturing of currently uncultured species. Moreover, such predictive model and the accompanied statistical analysis are of considerable importance for understanding the key aspects for community engineering and for the design of synthetic consortia. Though the text is generally well written, several parts are not sufficiently clear and additional information can be useful. Specific requests for clarifications are detailed below.

1. Tables showing the distribution of positive/negative interactions and the TP/TN/FP/FN for each category for the different methods used for the one-way interactions as well as similar information regarding the distribution of the different types of pairwise interactions will provide a clearer description of the data as well as a straightforward way of estimating how well the model performs considering different interaction types. Whereas the text analyses the effect of various features on prediction capacity, the effect of the type of interaction on model performances should also be discussed (at least the type/directionality of interactions).
2. Score of feature contribution to the model is not clearly explained. Each simulation has its own SHAPE value (Figure 3), but how was the feature's score determined? Also, the color bar is indicative of the SHAPE value of its simulation, however, it is located across the y axis which is confusing.
3. Models that were trained using information regarding each species' inferred metabolic pathways did not achieve higher prediction accuracy than models that used only phylogenetic information. This is explained by being inferred from the 16S sequences using picrust rather than being independent of phylogeny. However, metabolic PCs were inferred based on experimental performances rather than being based on phylogenetic data, yet with inferior performances. Do the metabolic PCs have an independent contribution to prediction quality?

Minor comments:

1. References 34 & 37 are redundant
2. L. 341. Should be "are given"
3. Figure 3: Phelogentic -> phylogenetic?
4. In general, figure quality is low (in particular the text).

The work by Nestor *et al.* uses a machine learning model to predict interspecies interactions from bacterial monoculture data, among other organism-specific traits. The authors leverage an extensive set of pairwise and monoculture data to characterize their model, as well as to test the contribution of different data sources on its performance. A thorough comparison with additional models is also presented. Overall, this is a clearly written and timely article that adds to the growing body of literature on machine learning methods for predicting microbiome behavior. Importantly, it also underscores how monoculture growth data is essential for correct predictions of interactions. There are a few points that in my view require further clarification, which are outlined below along with some suggestions on how the manuscript may be improved.

1. The authors use outputs from principal component analyses as features in their model (introduced in lines 77 and 349). I believe the authors are referring to Figure S9, though this figure is not referenced anywhere in the manuscript.
2. The authors contextualize their work by discussing the use of metabolic models (among other methods) and that these methods rely solely on genomic information (beginning at Line 41). While many studies indeed use models generated automatically from genome annotations, it is known that experimental curation is an important step in generating reliable quantitative predictions (PMID: 20057383), and has been applied extensively in model generation (PMID: 31391098, 28266498, 29692801, 21483480). As this is in line with the authors' observations on the essentiality of species monoculture growth, I would recommend rephrasing this statement.
3. Line 71: "in order to predict how species affect each other:" please specify what is being affected.
4. Line 73: Please change "growth" to "growth yield."
5. Line 79: Please change to "principal components."
6. Line 96: For a general audience, a description of tree-based models (and specifically of ensemble models like XGBoost) would be very useful.
7. Line 110: "most errors in sign prediction occurred for effects whose strength was close to 0." This is very interesting – were there inconsistencies in interaction sign for these examples in the original dataset? Are there other factors generally associated with these weaker interactions?
8. Line 139: Please provide citations for "which were previously shown to be predictive of interactions."
9. Figure 4: A calculation of statistical significance between the NMRSE distributions would be useful in more fully comparing each model's performance.

10. The model's predictions are based on species' growth yields in mono- and co-culture. While this is appropriate for this experimental system that contained limiting amounts of resources, natural microbiomes are open systems subject to nutrient cycling and periodic resource replenishment. I understand this study is framed as a proof of concept, but it would be valuable for the authors to comment on whether they might expect other ecologically relevant quantities (e.g., growth rate) to have similar degrees of predictive power in a non-laboratory setting. This could help guide the selection of features for future models intended for more complex microbiomes.
11. Line 335: For those not familiar with the original dataset, please clarify the degree to which each species was represented in the mono- and co-culture datasets.
12. Line 361: "highest accuracy for qualitative predictions" – is this appropriate given the imbalanced nature of the dataset (i.e. more negative interactions)?
13. Figure S2B: The total N should be specified or the figure should be normalized to the total number of effects.

We thank the reviewers for their comments. We have revised the manuscript based on these comments (changes are highlighted within the revised manuscript document). In addition, we have added a short paragraph to the discussion regarding limitations of our models concerning natural communities. Below we provide a detailed point-by-point reply to the Reviewers' comments:

Note: line numbers correspond to the pdf file.

Editor's comments:

Please add a brief discussion on the limitations of the approach when concerning natural communities whereby only a limited subset of species can be cultured, and potential artefacts of laboratory cultivations.

Thank you, this is indeed important to discuss. We have added a paragraph regarding the limitations of our work when considering natural communities:

"While our results demonstrate that bacterial interactions are predictable under simple laboratory conditions, it is still not clear to what extent this predictability extends to non-laboratory settings and to natural communities. First, many microbes are hard to isolate and culture in the lab (45, 46), and therefore monoculture growth yield will typically not be available for many environmental microbes. Since monoculture growth yield was the most informative feature for interaction predictions, not having this information would likely significantly reduce predictability. Moreover, the number of available resources natural communities are exposed to is larger than what our models were trained on, which were simple environments containing a single carbon source. This may make predicting interactions more challenging as species may occupy different niches and both grow well in monoculture without negatively influencing each other's growth. Lastly, our model predicts pairwise interactions, and does not account for the presence of "higher order" interactions (47). Therefore, even if our model accurately predicts the interactions between all species pairs of a natural community, the presence of additional species in the environment may modify these pairwise interactions". (Lines 353-366)

Reviewer #1 (Comments for the Author):

The work by Nestor et al. uses a machine learning model to predict interspecies interactions from bacterial monoculture data, among other organism-specific traits. The authors leverage an extensive set of pairwise and monoculture data to characterize their model, as well as to test the contribution of different data sources on its performance. A thorough comparison with additional models is also presented. Overall, this is a clearly written and timely article that adds to the growing body of literature on machine learning methods for predicting microbiome behavior. Importantly, it also underscores how monoculture growth data is essential for correct predictions of interactions. There are a few points that in my view require further clarification,

which are outlined below along with some suggestions on how the manuscript may be improved.

1. The authors use outputs from principal component analyses as features in their model (introduced in lines 77 and 349). I believe the authors are referring to Figure S9, though this figure is not referenced anywhere in the manuscript.

Thank you for pointing out this omission. We have added a reference to Figure S9 (Note that it is Figure S1 in the revised manuscript) in lines 98 and 399.

2. The authors contextualize their work by discussing the use of metabolic models (among other methods) and that these methods rely solely on genomic information (beginning at Line 41). While many studies indeed use models generated automatically from genome annotations, it is known that experimental curation is an important step in generating reliable quantitative predictions (PMID: 20057383), and has been applied extensively in model generation (PMID: 31391098, 28266498, 29692801, 21483480). As this is in line with the authors' observations on the essentiality of species monoculture growth, I would recommend rephrasing this statement.

Thank you for the detailed comment, this is an important clarification. We have rephrased the statement:

*"These approaches are appealing since they rely **primarily** on genomic information". (Line 61)*

In addition, we have added the following sentences to the discussion to emphasize the contribution of monoculture growth yields measurements to genome-based approaches:

*"The ability of the affected species to grow in monoculture in a given carbon environment was the feature that contributes the most to prediction. **This is consistent with the fact that methods such as metabolic modelling, which are primarily based on genomic information, require refinement using experimental measurements such as monoculture growth (38–42)**". (Lines 316-319)*

3. Line 71: "in order to predict how species affect each other:" please specify what is being affected.

We have rephrased this statement to clarify that species' growth is being affected:

*"In order to predict how species affect each other's **growth**, we have used additional information, beyond the interspecific interactions, regarding the species' phylogeny and their monoculture yield in each of the 40 carbon environments". (Lines 88-90)*

4. Line 73: Please change "growth" to "growth yield."

5. Line 79: Please change to "principal components."

Thank you, we made the changes suggested in both comments (4 + 5)

6. Line 96: For a general audience, a description of tree-based models (and specifically of ensemble models like XGBoost) would be very useful.

While a detailed introduction to tree-based models is beyond the scope of this manuscript, we agree that it is useful to mention the main concepts and ideas behind models such as XGBoost. To do so, we have added the following short description in the text:

“Briefly, XGboost uses an ensemble of decision trees, where each tree makes predictions by iteratively splitting the data based on individual features. XGboost implements a form of gradient boosting such that each new tree focuses on the samples where the previous trees had the highest error rates. XGboost is widely used since it provides accuracy and efficiency even on large datasets containing many features (34)”. (Lines 117-122)

7. Line 110: "most errors in sign prediction occurred for effects whose strength was close to 0." This is very interesting - were there inconsistencies in interaction sign for these examples in the original dataset? Are there other factors generally associated with these weaker interactions?

The true value of effect (that the model is trying to predict) is the median of all replicates of the same pair of species in the tested environment. We chose to classify interaction as negative if the effect is less than or equal to 0. However, there is often variability between replicates, and when the median effect is close to 0 there might be replicates with negative effect and ones with positive effect. Therefore, in these cases there is indeed some uncertainty about the true value of the effect that is being predicted.

Weak interactions are common between species that both grow very poorly in monoculture. In these cases, both species typically grow poorly both in monoculture and in coculture, making the measured effects more susceptible to measurement noise. Beyond that, we could not identify specific factors that are associated with these weaker interactions.

8. Line 139: Please provide citations for "which were previously shown to be predictive of interactions."

A citation was added to the sentence:

“Surprisingly, using information regarding species’ predicted metabolic pathways, which were previously shown to be predictive of interactions (28), instead of information regarding monoculture growth did not improve the predictive ability over a model that only used the species’ phylogeny (Fig. S6)”. (Lines 168-171)

28. DiMucci D, Kon M, Segrè D. 2018. Machine Learning Reveals Missing Edges and Putative Interaction Mechanisms in Microbial Ecosystem Networks. *mSystems* 3:e00181-18.

9. Figure 4: A calculation of statistical significance between the NRMSE distributions would be useful in more fully comparing each model's performance.

We have performed Tukey's HSD test and found that the increase in NRMSE due to the removal of monoculture data is indeed statistically significant. For the novel carbon environments (Fig. 4D), removing the coculture data results in a smaller, yet also statistically significant increase in the NRMSE. We added the results of Tukey's HSD to Figure 4, and included more detailed information regarding both tests in a new supplementary table (Table S3).

Figure 4. The accuracy of predicting one-way effect depends strongly on the availability of monoculture growth data, but not on coculture data. A. A summary of the 4 compared models and the information used by each model. Partial information (left to right): Coculture, Monoculture (of both species), phylogenetic tree and type of model (machine learning or simple

decision model). The models differ by the features they are trained on and their test set (see Methods). **B.** Comparison of NRMSE (normalized root mean square error) of several partial quantitative models, on “uncultured” strain. Each dot represents the NRMSE of a different strain that was excluded from the train set and was included in the test set. The models are the same as in A. Dashed line represents the performance of a null model (average effect in the train set). **C.** Comparison of NRMSE (normalized root mean square error) of several partial quantitative models, on “uncultured” carbon environment. Each dot represents the NRMSE of a different carbon source that was excluded from the train set and was included in the test set. The models are the same as in B. Dashed line represents the performance of a null model (average effect of train set). Qualitatively similar results were found also for predictions of effect sign (Fig. S7). **P-values for subplots B and C were calculated using Tukey-HSD test (see Table S3 at 10.6084/m9.figshare.21856578 for additional information regarding the p-values).** (Lines 230-245)

10. The model's predictions are based on species' growth yields in mono- and co-culture. While this is appropriate for this experimental system that contained limiting amounts of resources, natural microbiomes are open systems subject to nutrient cycling and periodic resource replenishment. I understand this study is framed as a proof of concept, but it would be valuable for the authors to comment on whether they might expect other ecologically relevant quantities (e.g., growth rate) to have similar degrees of predictive power in a non-laboratory setting. This could help guide the selection of features for future models intended for more complex microbiomes.

Thank you for the detailed comment, this is indeed important to discuss. As noted in our response to a similar comment from the Editor, we have added a paragraph regarding the limitations of our work when considering natural communities:

“While our results demonstrate that bacterial interactions are predictable under simple laboratory conditions, it is still not clear to what extent this predictability extends to non-laboratory settings and to natural communities. First, many microbes are hard to isolate and culture in the lab (45, 46), and therefore monoculture growth yield will typically not be available for many environmental microbes. Since monoculture growth yield was the most informative feature for interaction predictions, not having this information would likely significantly reduce predictability. Moreover, the number of available resources natural communities are exposed to is larger than what our models were trained on, which were simple environments containing a single carbon source. This may make predicting interactions more challenging as species may occupy different niches and both grow well in monoculture without negatively influencing each other’s growth. Lastly, our model predicts pairwise interactions, and does not account for the presence of “higher order” interactions (47). Therefore, even if our model accurately predicts the interactions between all species pairs of a natural community, the presence of additional species in the environment may modify these pairwise interactions”. (Lines 353-366)

The predictive power of machine learning models for other quantities, such as growth rate, is still unknown, but it would be very interesting to examine in the future. We speculate that,

similar to interactions, it will be possible to accurately predict additional ecological parameters in simple laboratory conditions, but that predicting these parameters in natural settings will face similar challenges as the ones discussed above for predicting interactions.

11. Line 335: For those not familiar with the original dataset, please clarify the degree to which each species was represented in the mono- and co-culture datasets.

For monoculture, the dataset included all the species grown in all carbon environments. The coculture data includes ~93% of all possible combinations of species and carbon environments. Combinations that were not included are those for which the original dataset included less than 4 replicates. We added these details to the method section, and additional information regarding the specific missing coculture data was added as supplementary text (Table S5):

“The dataset contains over 7500 pairwise interactions involving 20 species from 2 taxonomic groups in 40 different carbon environments (see previous work (33)), as well as the monoculture growth yield of all species in all carbon environments. ~7% of all possible combinations of species and carbon environments was not included in our analysis since they were represented by less than 3 replicates in the original experimental dataset (the specific missing combinations are listed in Table S4, see at 10.6084/m9.figshare.21856581)”. (Lines 379-384)

12. Line 361: "highest accuracy for qualitative predictions" - is this appropriate given the imbalanced nature of the dataset (i.e. more negative interactions)?

This is a very important comment, thank you. Indeed, all models should be hyper-tuned according to a score which is more appropriate to use when dealing with imbalanced data. We retrained all our models in order to maximize the MCC (Matthew's correlation coefficient) and update all results using the newly trained models. Two of threshold models (monoculture growth threshold and metabolic distance threshold) were retrained as well in order to maximize the MCC (and not the accuracy) on the train set. Using the MCC for cross-validation had only a small effect on model performance (MCC of the test set changed from 0.636 to 0.658). In addition, the type of the best-performing model (XGboost) didn't change. We have updated the manuscript accordingly.

13. Figure S2B: The total N should be specified or the figure should be normalized to the total number of effects.

We have added the total N to the figure (Note that this is Figure S3B in the revised manuscript).

Reviewer #2 (Comments for the Author):

The manuscript "Interactions between culturable bacteria are predicted by individual species growth" by Nestor et al aims at building a predictive model for pairwise interactions between bacteria. The predictive model is based on a rather unique collection of 7500 experimental

measurements of pairwise growth of 20 species grown on various carbon sources that was built as part of an earlier study (Kehe et al 2021). The current study is a natural continuation, using this data as a training set for a predictive model. The authors first demonstrate the performances of their model and then analyze the contribution of each of the different features on the performance. The importance of this work is in providing a guideline for the design of strategies for the culturing of currently uncultured species. Moreover, such predictive model and the accompanied statistical analysis are of considerable importance for understanding the key aspects for community engineering and for the design of synthetic consortia. Though the text is generally well written, several parts are not sufficiently clear and additional information can be useful. Specific requests for clarifications are detailed below.

1. Tables showing the distribution of positive/negative interactions and the TP/TN/FP/FN for each category for the different methods used for the one-way interactions as well as similar information regarding the distribution of the different types of pair-wise interactions will provide a clearer description of the data as well as a straightforward way of estimating how well the model performs considering different interaction types. Whereas the text analyses the effect of various features on prediction capacity, the effect of the type of interaction on model performances should also be discussed (at least the type/directionality of interactions)

Examining the effect of the interaction type on prediction accuracy is indeed an interesting point. Thank you for this suggestion. We found that positive effects are harder to predict and that mutualisms (+/+) are particularly challenging – they are more often classified as parasitisms (+/-) than as mutualisms. We have included these results in the main text:

“Moreover, it appears that predictions involving positive effects were less accurate: true negative effects were correctly classified more frequently than positive ones (95% true negative rate vs 66% true positive rate, Fig. S4A) and effects classified as negative were more likely to be true than effects classified as positive (90% negative predictive value vs 80% positive predictive value, Fig. S4A)”. (Lines 137-141)

“As positive one-way effects are harder to predict, mutualisms (+/+) are particularly challenging and are more often classified as parasitisms (+/-) than as mutualisms. (48% vs 33%, Fig. S4B).” (Lines 283-285)

Moreover, we have added a new supplementary figure (Figure S4 in the revised manuscript) showing the confusion matrices for the predictions of both one-way and two-way interactions:

Figure S4. Interactions involving positive effects are harder to predict. A. Confusion matrices for all examined machine learning models for sign predictions. **B.** Confusion matrix of the performance of two-way prediction. Predictions are made using one-way models (See methods). N = 3015 for both A and B.

2. Score of feature contribution to the model is not clearly explain. Each simulation has its own SHAP value (Figure 3), but how was the feature's score determined? Also, the color bar is indicative of the SHAP value of its simulation, however, it is located across the y axis which is confusing.

We apologize this wasn't clear enough. Each simulation is randomly changing the values of a single feature, and is given a SHAP value. The feature's score is the mean absolute SHAP value of all simulations that randomly changed this feature. The color bar indicates the value of the feature itself, and not the SHAP value. This enables us to connect between feature value and contribution to the model. We have added these clarifications in the text:

“Figure 3. Monoculture growth is the most important feature for predicting the strength of effect one species has on another’s growth. Top 10 most important features in the best performing strength prediction model (XGboost). High SHAP values indicate positive influence on the predicted effect, and low values indicate negative influence on the predicted effect. Y-axis represents the different features, sorted in descending order according to their contribution to the model which is defined by the mean absolute SHAP value of all simulations for that feature. Each dot represents a simulation of the model with a single change at a single feature value. Note that the different colors don’t represent the SHAP value, but the value of the feature. Similar contribution of monoculture growth features appeared also in sign predictions (Fig. S5A).” (Lines 174 - 182)

3. Models that were trained using information regarding each species' inferred metabolic pathways did not achieve higher prediction accuracy than models that used only phylogenetic information. This is explained by being inferred from the 16S sequences using picrust rather than being independent of phylogeny. However, metabolic PCs were inferred based on experimental performances rather than being based on phylogenetic data, yet with inferior performances. Do the metabolic PCs having independent contribution to predictions quality?

Thank you for this comment. To test the contribution of the metabolic PCs, we trained additional models based on the features used by the full sign and strength models except the metabolic PCs, which were removed. Overall, the NRMSE of the strength model increased from 0.36 to 0.38 and the MCC of the sign model decreased from 0.66 to 0.64. This indicates that the metabolic PCs do make an independent contribution to prediction quality, but this is a relatively small contribution. Indeed, this result is expected from the relatively low SHAP values of these features.

Minor comments:

1. References 34 & 37 are redundant
2. L. 341. Should be "are given"
3. Figure 3: Phelogentic -> phylogenetic?
4. In general, figures quality is low (in particular the text).

Thank you for finding these issues, we fixed all of them in the revised manuscript.

January 30, 2023

Dr. Jonathan Friedman
Hebrew University of Jerusalem
Rehovot
Israel

Re: mSystems00836-22R1 (Interactions between culturable bacteria are predicted by individual species' growth)

Dear Dr. Jonathan Friedman:

Please correct the typographical errors pointed out by the reviewer 1.

Your manuscript has been accepted, and I am forwarding it to the ASM Journals Department for publication. For your reference, ASM Journals' address is given below. Before it can be scheduled for publication, your manuscript will be checked by the mSystems production staff to make sure that all elements meet the technical requirements for publication. They will contact you if anything needs to be revised before copyediting and production can begin. Otherwise, you will be notified when your proofs are ready to be viewed.

If you would like to submit a potential Featured Image, please email a file and a short legend to msystems@asmusa.org. Please note that we can only consider images that (i) the authors created or own and (ii) have not been previously published. By submitting, you agree that the image can be used under the same terms as the published article. File requirements: square dimensions (4" x 4"), 300 dpi resolution, RGB colorspace, TIF file format.

We recognize that the video files can become quite large, and so to avoid quality loss ASM suggests sending the video file via <https://www.wetransfer.com/>. When you have a final version of the video and the still ready to share, please send it to mSystems staff at msystems@asmusa.org.

Sincerely,

Kiran Patil
Editor, mSystems

Journals Department
E-mail: mSystems@asmusa.org